Determining the influencing factors of preferential flow in ground fissures for coal mine dump eco-engineering

Li Yexin 1 2
Lv Gang 1
http://orcid.org/0000-0002-3921-5485 Shao Hongbo 3 4 5 shaohongbochu@126.com
Dai Quanhou 6
Du Xinpeng 1
Liang Dong 1
Kuang Shaoping 5
Wang Daohan 1 wangdaohan@sina.com
1 College of Environmental Science and Engineering, Liaoning Technical University , Fuxin , P.R. China
2 School of Architecture and Civil Engineering, Shenyang University of Technology , Shenyang , P.R. China
3 Salt-Soil Agricultural Center, Key Laboratory of Agricultural Environment in the Lower Reaches of Yangtze River Plain, Institute of Agriculture Resources and Environment, Jiangsu Academy of Agricultural Sciences (JAAS) , Nanjing , P.R. China
4 Jiangsu Key Laboratory for Bioresources of Saline Soils, Jiangsu Synthetic Innovation Center for Coastal Bio-agriculture, Yancheng Teachers University , Yancheng , P.R. China
5 College of Environment and Safety Engineering, Qingdao University of Science & Technology(QUST) , Qingdao , P.R. China
6 College of Forestry, Guizhou University , Guiyang , P.R. China
Scheibe Timothy
Electronic publication date: 2021 Jan 5
Publication date: 2021
Volume: 9
Electronic Location ID: e10547
Received 2020 Aug 31; Accepted 2020 Nov 20
Copyright: © 2021 Li et al.
Copyright year: 2021
Copyright holder: Li et al.
License: This is an open access article distributed under the terms of the Creative Commons Attribution License, which permits unrestricted use, distribution, reproduction and adaptation in any medium and for any purpose provided that it is properly attributed. For attribution, the original author(s), title, publication source (PeerJ) and either DOI or URL of the article must be cited.
License URL: https://creativecommons.org/licenses/by/4.0/

Keywords: Preferential flow, Eco-engineering dump, Dye tracer, Ground fissures, Roots

Funding: National Key Research and Development Program of China 2017YFC1503105 Excellent Scientist Plan of JAAS, China This study was financially supported by the projects of the National Key Research and Development Program of China (2017YFC1503105) and Excellent Scientist Plan of JAAS, China. The funders had no role in study design, data collection and analysis, decision to publish, or preparation of the manuscript.

==============================
Ground fissures (GF), appearing in front of dumps, are one of the most obvious and harmful geological hazards in coal mining areas. Studying preferential flow and its influencing factors in the ground fissures of dumps may provide basic scientific support for understanding the rapid movement of water and vegetation restoration and reconstruction in mining areas. Based on field surveys of ground fissures, three typical ground fissures were selected in the studied dump. The morphological characteristics of preferential flow for ground fissures were determined through field dye tracing, laboratory experiments, and image processing technology. The results indicated that the lengths of the three ground fissures ranged from 104.84 cm to 120.83 cm, and the widths ranged from 2.86 cm to 9.85 cm. All of the ground fissure area densities were less than 10%, and the proportion of ground fissure surface area was small in the dump. The maximum fissure depth was 47 cm, and the minimum was 16 cm. The ground fissure widths ranged from 0 cm to 14.98 cm, and the fissure width and fissure width-to-depth ratios decreased with increasing soil depth. The stained area was greater than 90% in the 0–5 cm soil layers of the three fissures, and water movement was dominated by matrix flow. The stained width decreased from 90 cm to 20 cm with increasing soil depth. The preferential flow was mainly concentrated on both sides of the fissure, which was distributed as a “T” shape. The preferential flow stained area ratios were 27.23%, 31.97%, and 30.73%, respectively, and these values decreased with increasing soil depth. The maximum stained depths of the preferential flow among the three fissures were different, and the maximum stained depth of GF II was significantly larger than that of GF I and GF III (P < 0.05). The stained path numbers of the three fissures ranged from 0 to 49. With increasing soil depth, the stained path number first increased and then decreased. The stained path widths of the three fissures ranged from 0 cm to 90 cm. With the increase in soil depth, the stained path width decreased. The stained area ratio was significantly positively correlated with ground fissure width, the ground fissure width-to-depth ratio, soil saturated hydraulic conductivity, soil organic matter, and sand content and was significantly negatively correlated with soil water content and clay content. The stained path number was significantly positively correlated with ground fissure width, the ground fissure width-to-depth ratio, soil saturated hydraulic conductivity and soil organic matter. The stained path width was significantly positively correlated with the ground fissure width-to-depth ratio, soil saturated hydraulic conductivity, soil organic matter and sand content and was significantly negatively correlated with clay content. Plant roots could significantly increase the stained area ratio, stained path number, and width and promote the formation and development of preferential flow.

Introduction

Global climate change can lead to frequent regional droughts and floods and uneven distributions of precipitation and water resources. Global climate change can also gradually increase the frequency and intensity of extreme weather events, causing serious damage to agricultural production. With the occurrence of droughts, soil moisture decreases continuously under evaporation and dry conditions, resulting in different degrees of cracks in the earth and ground fissures (Gaur, Kar & Srivastava, 2015; Ren et al., 2016). Ground fissures are affected by different factors, such as soil texture, soil organic carbon (Kechavarzi, Dawson & Leedsharrison, 2010), alternating soil drying and wetting (Zhang et al., 2013), freeze and thaw action (Maloof, Kellogg & Anders, 2002), and tillage methods (Bandyopadhyay et al., 2003; Zhang et al., 2016b). The formation of ground fissures not only changes the soil structure (Bruand, Cochrane & Fisher, 2001) and increases the infiltration path and capacity of surface runoff (Liu et al., 2003) but also affects the growth, development, and ecohydrological processes of vegetation, leading to land degradation. Land degradation is a global problem that has attracted substantial attention from scholars (García-González et al., 2018; Nabiollahi et al., 2018; Shao et al., 2018; Zika & Erb, 2009).

Preferential flow is defined as a phenomenon in which soil water moves along certain pathways, such as soil macropores, soil cracks, wormholes, and plant roots, bypassing most of the porous matrix and quickly moving through the soil media (Bero, Ruark & Lowery, 2016; Hagedorn & Bundt, 2002; Hardie et al., 2011; Rye & Smettem, 2017). Soil preferential flow is caused by the spatial heterogeneity of the soil structure, which exists in the process of soil water infiltration (Clothier, Green & Deurer, 2008). Surroundings and imbalance are two important features of preferential flow (Zhang et al., 2016a). The soil hydrological process, with preferential flow as one of the factors influencing environmental issues, specifically land degradation and groundwater resource security, occurs throughout the world and has been studied at different scales from the field plot and hillslope scale to the catchment scale (Keesstra et al., 2016; Zehe et al., 2010). At present, many experts and scholars mainly focus on soil preferential flow characteristics and their influencing factors under different land uses (Allaire, Roulier & Cessna, 2009; Clark & Zipper, 2016; Laine-Kaulio et al., 2015; Leuther et al., 2018). Berli et al. (2004) carried out a study on agricultural and forest soils and found that soil with different compaction degrees has different preferential flow characteristics. Li & Ghodrati (1994) concluded that plant roots were the key factor affecting the preferential flow of forest soil. Liu & Du (2013) investigated the characteristics of soil preferential flows in different vertical traps in mountains along the Dalaoling–Dengcun section of the Three Gorges of the Yangtze River. The preferential flow paths in deserted farmland differ from those in forest soils, and the water infiltration intensity was low and the color of the stained area was light. Wu et al. (2014) investigated and quantified the preferential flow in five intact and five disturbed soil columns sampled by a Brilliant Blue FCF dye tracer experiment and noted that preferential flow developed in the intact soil columns, while piston flow developed in the disturbed soil columns. Preferential flow may accelerate or delay the movement of matter depending upon the position of the matter compared to the position of preferential flow paths. Four types of preferential flow can be considered: crack flow, finger flow, lateral flow and macropore flow (Allaire, Roulier & Cessna, 2009; Zhang & Peng, 2015).

Mine spoils are an important driver of environmental damage and land degradation, causing the disappearance of soil and vegetation; increases in erosion by wind and water; pollution of air, soil, and water; and general deterioration of the landscape (Mukaro, Nyakudya & Jimu, 2017). Coal mining will result in different degrees of surface damage and deformation, causing ground fissures (Yang et al., 2018; Zhang et al., 2015). Ground fissures have no obvious frame structure. The numbers of fissures, fissure blocks, and nodes are relatively small. The depths of the fissures are also different. Ground fissures have relatively strong spatial variability yet clear self-similarity (Díaz-Fernández, Álvarez-Fernández & Álvarez-Vigil, 2010; Li et al., 2018). These factors lead to different preferential flows (Guo et al., 2018). Cheng (2016) determined that the preferential flow of fissures was a direct factor in the change in the soil environment in a mining area and a driving factor of soil erosion in a mining area, and soil crevice preferential flow provides a preferential path for water to transport soil particles from the surface layer to the deep layer, and crevices with soil particles filled will become long-term preferential flow paths. Yan et al. (2018) carried out a study on ammonia concentration and particle size change in soil during the rainfall-runoff process and found that ground fissures with uneven settlement not only changed the surface topographic gradient and soil structure significantly but also aggravated the transport intensity of nutrients and particles in soil with runoff in the horizontal direction. Moreover, the authors determined that the soil cracks produced by surface subsidence formed a series of groundwater preferential flow paths and promote nutrient transport towards a deeper layer. Ground fissure preferential flow was the important driving factor for soil erosion. The subsidence area without any vegetation in Shenfu-Dongsheng coalfield has been taken as research area. Two test points were set in a 50 m × 50 m testing area, then 4 g/L bright-blue solution was used to dye it, and images were taken in 1 horizontal and 5 vertical sections, and the characteristic of preferential flow in the soil cracks of a coal mining subsidence area (Guo et al., 2018). In addition, soil cracks accelerate water infiltration into deep soil along cracks, reduce soil shear strength, raise groundwater levels, and induce natural disasters such as landslides and debris flows (Krzeminska et al., 2012; Kukemilks et al., 2017; Ma, 2017; Woerden, 2014). Reclaimed mine soil is an important manifestation of land degradation, which is caused by vegetation degradation, pressure station land and serious soil erosion. Therefore, understanding the land degradation response to preferential flow is essential.

In summary, studies of soil fissures in coal mining areas are mostly concentrated in subsidence areas, while few studies have focused on ground fissures in dump areas. Zhou et al. (2011) pointed out that rainfall and groundwater discharge was important factors for the formation and development of ground fissure. Wang et al. (2012) carried out that ground fissure were the precursor of landslide or debris flow hazard in the dump area. Moreover, the study of preferential flow in ground fissures is still lacking, especially in surface coal mine dumps. Ground fissure will change the surface runoff movement, causing a large amount of water to move to the deep soil in the dump area, which is easy to induce disasters, such as landslide and debris flow. A comprehensive understanding of the characteristics of preferential flow in the ground fissures of a dump is of great scientific significance for understanding water and soil loss and for implementing vegetation restoration in mining areas. The aims of this study were to (1) reveal the characteristics of the preferential flow of ground fissures and (2) clarify the influencing factors of soil preferential flow. The research results could provide a scientific basis for understanding the rapid movement of water, vegetation restoration, and eco-engineering in mining areas.

Methods and materials

Study site

The study area is in the south dump of Shenglidong Open-pit Coal Mine No. 2 of Datang International Power Generation Company in Xilinhot city, Xilingol league, Inner Mongolia (Fig. 1). The study area is located at 116°06′41″ to 116°14′11″ E and 44°02′0″ to 44°07′05″ N in the southeastern part of the mining area. The total area is 13.66 km2. The local climate is arid and semiarid in the middle temperate zone. The average annual temperature is 1.7 °C, and the average annual precipitation is 284.74 mm. Precipitation mainly occurs in June–August, accounting for more than 71% of the annual rainfall. Heavy rainfall usually and frequently occurs in these 3 months, especially between mid-July and mid-August. The long-term mean maximum precipitation in 24 h is 46.8 mm. The annual average evaporation is 1,794.6 mm, and the annual average wind speed is 3.4 m s−1. The freezing period is from early October to early December, while the thawing period is from late March to the middle of April. The maximum depth of frozen soil is 2.89 m, and the soil is typical chestnut soil. The climate conditions are mainly derived from China Meteorological science data sharing service platform, with an average value of 30 years. One dump is located to the south of the first mining zone and within the mining area; that is, the south and north dumps are set to the south and north of the first mining zone of the mining site, respectively. The service life of each dump is 20 years. The main design parameters of the south dump are shown in Table 1. To restore the vegetation of the dump as soon as possible, soil covering measures were applied for the platform and slope reclamation (the soil is sandy loam). The thickness of the covering soil on the platform is approximately 1 m and that over the slope is approximately 0.5 m. The vegetation for the reclamation is shrubs or herbage, such as Caragana korshinskii Kom., Hippophae rhamnoides Linn., Astragalus adsurgens Pall., Melilotus suaveolens Ledeb., and Medicago Sativa Linn.

Figure 1 Location of the study area.

(A) Location of the study area in Inner Mongolia of China. (B) Platform and slope of coal mine dump. (C) Ground fissures in the coal mine dump.

Table 1 Main design parameters of dump.

Area
(km2)	Final dumping
elevation (m)	Height (m)	Bench height
(m)	Platform width
(m)	Capacity
(Mm3)	Stepped slope
angle (°)	Loose
coefficient	
7.60	1,156	100	25	20	592	33	1.15	

Research methods

Sample selection and layout

Relevant studies show that the subsidence coefficient of a dump generally ranges between 1.1 and 1.2. The subsiding process lasts for several years, but the subsidence in the first 3 years can account for 80% of the total subsidence (Han, Bai & Li, 2011). In this study, Platform 1105 was taken as the research object. Soil covering was performed on Platform 1105 in 2013 after the platform was fully dumped (4 years dumping time). In August 2017, the distribution characteristics and development degree of the ground fissures on the platform were fully examined. According to the investigation, there were 61 ground fissures with different sizes on Platform 1105. The fissures stretched along the contour lines (the margins of the dump). Straight lines were the dominant form of the fissures. The fissures were concentrated within the first 5 m from the frontal edge of the dump. The surface parts of some fissures had cut through the platform and became interwoven with each other, forming a fissure zone. In fissure zones and areas with extensively developed ground fissures, collapse and breakoff often occur at the edges of fissures. All the ground fissure were divided into several groups according to their width, and perform mathematical statistical analysis to screen out the 3 groups with the highest frequency, and a typical ground fissure was selected in each group. Three typical fissures, which were marked GF I, GF II, and GF III, were selected as the study objects by statistical analysis. The horizontal morphological characteristics of the ground fissures were investigated by using “frame photography” methods (Han, Bai & Li, 2011; Kishné et al., 2010). At the same time, three soil profiles (30, 50 and 70 cm in the sample plot) were excavated in each sample plot to obtain the vertical distribution map of ground fissures. ArcGIS 10.5 and AutoCAD 2008 software were used to digitalize the photographs of the ground fissures and obtain the horizontal and vertical morphological characteristics. The fissure length is the length of the centerline of a fissure. The fissure width is the surface width of a fissure. The fissure depth is the vertical depth of a fissure. The fissure perimeter is the total length of the outline of a fissure, and the fissure area is the area within the outline of a fissure. The fissure length density is the ratio between the fissure length and sample square (100 cm × 100 cm). The fissure area density is the ratio between the fissure area and sample square (100 cm × 100 cm). The fissure width-to-depth ratio is the ratio between fissure width and fissure depth.

Dye-staining experiment and sample collection

In each experimental plot, four iron plates were embedded into the soil. The length and width of the iron plate were 105 cm and 50 cm, respectively. The rectangular iron frame was elevated 45 cm above the soil surface and 5 cm under the soil surface. To improve the test accuracy, the soil within 5 cm of the inside frame could be compacted. According to previous research results, preferential flow mostly occurred under heavy rainfall conditions. It was calculated that 50 L bright blue solution was needed for one test based on the size of the sample plot (105 cm × 105 cm) in terms of actual consumption and loss. The rainfall intensity at this time was 50 mm h−1. Brilliant Blue FCF dye was used in this study due to its good visibility in soils, low toxicity, low sorption, and transport properties similar to water (Germán-Heins & Flury, 2000). In each plot, 50 L of the aqueous solution of Brilliant Blue FCF (concentration = 4 g L−1) was applied over 1 h in the plot at a constant rate of 50 mm h−1 using a rainfall simulator, and then, a plastic cloth was placed around the iron plates to prevent water evaporation. The canvas was removed 24 h into the dye-staining experiment, and the section perpendicular to the ground fissure was selected as the observation surface of the dyeing test. Three soil profiles were laid at 30, 50 and 70 cm. The size was determined by placing the pole on the surface of the section in the form of a square. Then, a vertical acquisition dyeing profile was obtained using a digital camera with 12 million pixels after dressing the soil profile, and 3–5 photos were selected for the next picture process (Fig. 2). Stained images were collected at the same time, usually at 11 am. We can ensure that all tasks were completed before 11 am, except for stained image collection. At the same time, use PVC panels to block sunlight while taking pictures to avoid shadows. At the same time, along the vertical direction of the fissure, soil samples were measured at 10 cm intervals at a depth of 0–60 cm by a ring knife (5 cm high, 200 cm3 volume). The soil water content was measured using the drying method, and the soil bulk density, porosity (total porosity, capillary porosity, and non-capillary porosity), and saturated hydraulic conductivity were measured using the ring method (Shi et al., 2016). Then, 2 kg of soil sample was collected in each soil layer and taken to the laboratory for further analysis. The soil organic matter was measured using the potassium dichromate-external heating method. The soil mechanical composition was measured by the pipette method, and soil texture classification was carried out according to international classifications (FAO and USDA) (Institute of Soil Science of Chinese Academy of Sciences, 1978). All experiments were replicated three times. The roots were collected by using a root drilling method (10 cm diameter and 10 cm length) in each plot, and the sampling depth was the same as that of the soil samples. The root samples were washed from root drilling in the laboratory. The root length, surface area and volume were determined using WinRHIZO PRO 2009 software. The root biomass was weighed after being oven-dried at 75 °C.

Figure 2 The dyeing test process and image processing.

(A) Plot layout. (B) Spray brilliant blue FCF. (C) Sample plot excavation. (D) Soil profile. (E) Image correction. (F) Noise reduction.

Image processing

Four steps were necessary to obtain stained image characteristics, such as the stained area ratio, maximum stained depth, stained path number, and width. The first step was to correct the pictures of certain profiles that could not be photographed orthogonally by ERDAS IMAUINE 9.2 software and output JPG files of the RGB format with a size of 900 × 700 pixels. The second step was to adjust the new correct image using the image editing program Adobe Photoshop CS6. The third step was to convert the image from RGB format to grayscale images, denoise the image using Image-Pro Plus 6.0 software, and then output the TIFF files. One pixel represented an area of 1 mm × 1 mm in the true coordinate system. The stained areas turned black, while the remaining areas stayed white. A “0” was the expression of black, while “225” was the expression of white. The fourth step was to draw a numerical matrix of the bitmap with a 900 × 700 resolution and save it as an Excel file.

Index of soil preferential flow

Maximum stained depth

The maximum stained depth is the maximum stained depth of preferential flow in the soil profile, and this depth can most directly reflect the characteristics of rapid water movement and migration depth.

Stained path number and stained path width

The stained path number and width can reflect the connectivity, interaction and variability in preferential flow paths. The number of continuous black grids in each layer (1 cm) is calculated and recorded as a stained path. The number of stained paths in each layer is counted, and the actual width of the stained path is obtained by multiplying the proportion of one stained path by the actual width.

Stained area ratio and its coefficient of variation

The stained area ratio is the proportion of preferential flow stained area in a soil profile area, which is derived from stained pixels and unstained pixels in a soil profile. It not only reflects the proportion of stained area, but also reflects the distribution characteristics of soil preferential flow and its changes with soil depth. The stained area ratio has different characteristics because of the different width and depth of the ground fissures. It can be calculated from Eq. (1):

(1) K=PP+Q×100%

where K is the stained area ratio (%), P is the stained pixels in the soil profile (cm2), and Q is the unstained pixels in the soil profile (cm2).

The coefficient of variation of the stained area ratio can directly reflect the changes in the stained area at different depths of the soil profile. The smaller the coefficient is, the higher the degree of development of soil preferential flow. The coefficient of variation can be calculated from Eq. (2):

(2) Cv=Sx×100%=1n−1∑i=1n(xi−x)21n∑i=1nxi×100%

where Cv is the coefficient of variation of the stained area ratio (%), S is the standard deviation of the stained area ratio (%), x is the average of the stained area ratio (%), xi is the stained area ratio at i cm of soil profile (%), and n is the equipartition number of the stained image (each size is 1 cm).

Data analysis

Statistical data analysis was performed using Microsoft Excel 2003 and SPSS Statistics 17.0 software. The mean, standard deviation and coefficient of variation were calculated by Microsoft Excel 2003. Drawing was performed using Origin 8.0 software. Image processing was performed using ERDAS IMAUINE 9.2 software, Adobe Photoshop CS6 and Image-Pro plus 6.0 software. The least significant difference (LSD) method was used for multiple comparisons by SPSS Statistics 17.0 software. The LSD test uses the square root of the residual mean square from the one-way analysis of variance (ANOVA) and considers it to be the pooled significant difference.

Results

Characteristics of ground fissures in the studied dump

Table 2 shows the morphological characteristics of ground fissures in the studied dump. The lengths of the three ground fissures ranged from 104.84 cm to 120.83 cm, and all larger were than 100 cm (the side length of the sample square), indicating that the fissures had different degrees of curvature. Fissure width reflects the degree of opening and fissure disturbance to soil continuity. The ground fissure widths of GF I, GF II, and GF III were 9.85, 2.86, and 5.77 cm, respectively. There were significant differences among the three fissures (P < 0.05). Of the fissures, GF I had the greatest degree of opening and caused the most disturbance to soil continuity. The ground fissure area densities of GF I, GF II, and GF III were 9.84%, 2.86%, and 5.76%, respectively. There were significant differences among the three fissures (P < 0.05). All were less than 10%, which showed that the proportion of ground fissure surface area was small in the dump.

Table 2 Morphological characteristics of ground fissures in the dump.

Plots	Length
(cm)	Width
(cm)	Depth
(cm)	Perimeter
(cm)	Area
(cm2)	Length density
(cm cm−2)	Area density
(%)	
GF I	120.83	9.85 ± 3.48a	29 ± 6.08a	287.65	983.92a	0.012	9.84a	
GF II	109.90	2.86 ± 0.84c	30.67 ± 5.03a	230.65	286.47c	0.011	2.86c	
GF III	104.84	5.77 ± 2.76b	28.67 ± 15.82a	228.58	576.70b	0.010	5.76b	
Note:

Values are mean ± standard deviation. The width and depth were an average value of multiple measurement points. Different lowercase letters indicate significant differences between different plots (P < 0.05).

The vertical distribution of a ground fissure reflects the variation in fissure width with soil depth. As shown in Fig. 3, the ground fissure depths of GF I, GF II, and GF III were 29 cm, 30.67 cm, and 28.67 cm, respectively. There was no significant difference among the three ground fissures (P > 0.05). The maximum fissure depth of GF III was 47 cm, and the minimum fissure depth was 16 cm. A ground fissure has obvious spatial variability. The ground fissure width decreased with increasing soil depth. The fissure width of GF I ranged from 1.8 cm to 13.77 cm, that of GF II ranged from 0 cm to 10.58 cm, and that of GF III ranged mainly from 1.12 cm to 14.98 cm. The maximum fissure width appeared on the surface (soil depth was 0 cm), which indicates that disturbance to soil continuity decreased with increasing soil depth. The fissure width of GF I-70 cm decreased sharply from 8.61 cm to 1.8 cm at a depth of 19–22 cm. For GF II, the fissure widths of GF II-30 cm, GF II-50 cm, and GF II-70 cm decreased continuously at depths of 0–5 cm, and then, those of GF II-30 cm and GF II-50 cm fluctuated. The fissure width of GF II-70 cm was 0 at depths of 10–27 cm because the fissures in this position were buried by soil particles. The fissure widths of GF III-30 cm and GF III-50 cm decreased with increasing soil depth, and those of GF III-70 cm decreased sharply from 5.07 cm to 7.84 cm at 27 cm, showing a decreasing trend.

Figure 3 Change of ground fissure width with soil layer depth.

(A) The width of GF I. (B) The width of GF II. (C) The width of GF III.The red lines represent soil profiles with 30 cm, the green lines represent soil profiles with 50 cm, and the blue lines represent soil profiles with 70 cm.

Figure 4 indicates that as fissure depth increased, the ground fissure width-to-depth ratio significantly decreased. The ground fissure width-to-depth ratio decreased rapidly within the depth range of 0 cm to 10 cm and then stabilized. The width-to-depth ratios of different profiles in the same fissure had the same variation tendency, but the maximum width-to-depth ratio was different. The maximum width-to-depth ratios of the three soil profiles with GF I were 4.78, 8.97, and 13.77; those of GF II were 8.72, 10.14, and 10.58; and those of GF III were 10.33, 14.98, and 5.14. The stability of the width-to-depth ratios was only from 0.43% to 1.71% of the maximum value.

Figure 4 Ground fissure width-to-depth ratio in the dump.

(A) The ground fissure width-to-depth ratio of GF I. (B) The ground fissure width-to-depth ratio of GF II. (C) The ground fissure width-to-depth ratio of GF III.The red points represent soil profiles with 30 cm, the green points represent soil profiles with 50 cm, and the blue points represent soil profiles with 70 cm.

Characteristics of preferential flow in studied the dump

Morphological characteristics of preferential flow

Figure 5 shows the vertical distribution characteristics of preferential flow for the three ground fissures. The stained area was greater than 90% in the 0–5 cm soil layer for the three fissures. The stained area was evenly distributed, and preferential flow development was not obvious. Water movement was dominated by matrix flow, which was due to the effect of matrix flow being greater than that of preferential flow. The stained width decreased from 90 cm to 20 cm with increasing soil depth. The preferential flow stained area was mainly concentrated on both sides of the fissure and was closely related to the preferential movement of water to the ground fissure. The preferential flow was mainly concentrated in the middle of the stained image, which was distributed as a “T” shape, but there were no significant differences for GF I, GF II and GF III.

Figure 5 Vertical distribution of preferential flow.

(A) The soil profile of 30 cm in GF I. (B) The soil profile of 50 cm in GF I. (C) The soil profile of 70 cm in GF I. (D) The soil profile of 30 cm in GF II. (E) The soil profile of 50 cm in GF II. (F) The soil profile of 70 cm in GF II. (G) The soil profile of 30 cm in GF III. (H) The soil profile of 50 cm in GF III. (I) The soil profile of 70 cm in GF III.

Stained area ratio

The stained area ratios of GF I, GF II and GF III were 27.23%, 31.97% and 30.73%, respectively, which decreased with increasing soil depth and were distributed in an “S” shape (Fig. 6). The stained area ratio was the largest in the 0–10 cm soil layer, and the ratios of GF I, GF III and GF III was 70.97–82.92%, 62.51–90.70% and 70.34–89.06%, respectively. In the 0–10 cm soil layer, the stained area ratio decreased rapidly. The stained area ratios of GF I, GF II and GF III in the 0–1 cm soil layer were 99.93%, 95.57%, 99.98%, respectively, whereas those at 10 cm were 39.72%, 42.13%, 42.79%. The stained area ratio decreased with depth; however, this decrease was not linear but fluctuated. When the soil depth was 60–70 cm, the stained area ratio tended to be 0.

Figure 6 Stained area ratio with soil depth.

(A & B) The stained area ratio of GF I. (C & D) The stained area ratio of GF II. (E & F) The stained area ratio of GF III. The black lines and black column represent soil profiles with 30 cm, the pink lines and pink column represent soil profiles with 50 cm, and the blue lines and blue column represent soil profiles with 70 cm.

As shown in Fig. 6, the maximum stained depth of the preferential flow among the three fissures was different; the 3 soil profiles for GF I were 50 cm, 55.6 cm, and 54.9 cm; those for GF II were 65 cm, 63.6 cm, and 59.7 cm; and those for GF III were 56.5 cm, 55.6 cm, and 53.9 cm. GF II was significantly larger than GF I and GF III (P < 0.05), and this result was related to the fissure width and depth of GF II. In the 0–70 cm soil layer, the maximum stained area ratios of the three fissures appeared in the 0–1 cm soil layer, and the values of GF I, GF II, and GF III were 99.93%, 95.57%, 99.98%, respectively. The minimum stained area ratio was 0, which appeared at 60–70 cm. For GF I, GF II and GF III, the average values of the stained area ratio were 27.23%, 31.97%, and 30.72%, respectively. The results above showed that the stained area ratio was highly discrete and variable. The coefficient of variation for GF I was 78.95–95.24%, that of GF II was 68.45–77.03%, and that of GF III was 62.92–87.67%.

Stained path number and width

Figure 7 shows the stained path number and width of three ground fissures. As shown, the stained path number of the three fissures ranged from 0 to 49. With increasing soil depth, the stained path number first increased and then decreased. The maximum stained path number among the three fissures appeared at different soil depths. For the three soil profiles of GF I, the maximum stained path numbers were 20, 36 and 38, which appeared at depths of 8, 7 and 6 cm, respectively. For the three soil profiles of GF II, the maximum path numbers were 35, 21 and 18, which appeared at depths of 3, 3 and 18 cm, respectively. For the three soil profiles of GF III, the maximum path numbers were 43, 33 and 49, which appeared at depths of 7, 10 and 4 cm, respectively. The maximum stained path numbers of all the soil profiles appeared in the 0–10 cm soil layer, except that GF II-70 cm was at 27 cm. This result showed that the water flow morphological differentiation was the most serious in the surface soil. The stained path number decreased below the 10 cm soil layer, and the value was 0–23. However, this decrease was not linear but fluctuated. The stained path was mainly concentrated on the area directly below the fissure, which was closely related to the water catchment capacity of the fissure.

Figure 7 Distribution of stained path number.

(A) The stained path number of GF I. (B) The stained path number of GF II. (C) The stained path number of GF III. The red lines represent soil profiles with 30 cm, the green lines represent soil profiles with 50 cm, and the blue lines represent soil profiles with 70 cm.

Table 3 shows the stained path number and width in different soil layers. With the increase in soil depth, the stained path number of all soil profiles decreased first and then increased, and the maximum appeared in the 0–10 cm soil layer. However, the stained path number fluctuated greatly at 0–10 cm, with a maximum of 49 and a minimum of 1. For the stained area with only one stained path, the stained path width was 90 cm (the stained image width was 90 cm), and the stained area covered the whole soil profile. This result showed that this area was a matrix flow area. With the increase in soil depth, the stained path width decreased and was only 9.1–41 cm in the 10–20 cm soil layer, accounting for 45.59% of the 0–10 cm soil layer. The stained path width of some soil profiles in the 50–60 cm soil layer was 0, and there was no stained area. In the 60–70 cm soil layer, only the stained path widths of GF II-30 cm and GF II-50 cm were not 0, and the values were 5.2 cm and 4.8 cm, accounting for 5.78% and 5.33% of the stained image width, respectively. The stained area was not obvious, and the preferential flow reached the maximum depth.

Table 3 Stained path number and width with different soil depth.

Stained path	Soil depth
(cm)	GF I	GF II	GF III	
30 cm	50 cm	70 cm	30 cm	50 cm	70 cm	30 cm	50 cm	70 cm	
Number	0–10	4~20	1~36	1~38	6~35	1~21	1~14	1~43	1~33	2~49	
10–20	1~6	2~10	2~9	6~20	3~15	7~13	4~20	4~32	9~21	
20–30	1~7	1~6	1~5	2~19	1~9	8~18	4~13	2~8	1~23	
30–40	2~12	1~3	1~5	1~5	1~4	2~14	4~15	4~13	2~5	
40–50	0~7	1~5	3~9	1~4	1~5	3~8	2~5	3~12	1~3	
50–60	0	0~9	0~4	1~6	1~10	0~8	0~6	0~7	0~2	
60–70	0	0	0	0~3	0~13	0	0	0	0	
Width
(cm)	0–10	30.5~89	22.8~90	21.5~90	21.4~67.3	28.9~89.3	44.6~90	21.9~90	28.2~90	26~89.8	
10–20	12.5~21	14.8~21.8	17.9~22	14.9~26.3	18.2~31.7	12.2~41	18~20.8	9.1~26.7	17.1~29.1	
20–30	15.8~23.4	17.9~26.6	18.3~22.8	2.8~19.1	13.4~20	15.2~24.4	13.8~21.1	7.7~12.3	13.2~31	
30–40	6.7~18.5	14.3~20.5	15.4~21.2	5.3~18.9	17.4~38	11.1~30.6	11.9~23.2	10~25.8	19.6~25.2	
40–50	0~6.2	11.1~17	15.3~27	19.8~23.9	8.5~24.1	12.8~24.7	12.3~24.1	11.9~25.9	8.7~22.9	
50–60	0	0~12.1	0~25.7	10~17.7	6.8~9.5	0~25.3	0~12.3	0~8.7	0~9.1	
60–70	0	0	0	0~5.2	0~4.8	0	0	0	0	

Influencing factors of soil preferential flow in the studied dump

Ground fissure

As shown in Table 4, the stained area ratio and stained path width were significantly positively correlated with ground fissure width (P < 0.01), and the stained path number was significantly negatively correlated with ground fissure width. The increase in ground fissures significantly increased the stained area ratio and stained path width, providing a preferential channel for water movement. The stained area ratio and stained path width were significantly positively correlated with the ground fissure width-to-depth ratio (P < 0.01), and the stained path number was significantly positively correlated with the ground fissure width-to-depth ratio (P < 0.05). The results above showed that preferential flow morphological characteristics were affected by both ground fissure width and depth. The wider and deeper the ground fissure was, the greater the influence was, and the more obvious the preferential flow development was.

Table 4 Relationship between soil preferential flow and ground fissure width and width-to-depth ratio.

Index	SAR	SPN	SPW	GFW	GFWR	
SAR	1.000	0.336**	0.947**	0.309**	0.681**	
SPN		1.000	0.168**	−0.107	0.123*	
SPW			1.000	0.346**	0.649**	
GFW				1.000	0.746**	
GFWR					1.000	
Notes:

* P < 0.05.

** P < 0.01.

n = 270. SAR, stained area ratio; SPN, stained path number; SPW, stained path width; GFW, ground fissure width; GFWR, ground fissure width-to-depth ratio.

Soil physiochemical properties

Table 5 shows the soil physicochemical properties of the ground fissures in the studied dump. For the three ground fissures, the contents of sand, silt, and clay for the different soil layers were 58–78.62%, 10.25–17.64% and 6.72–14.63%, respectively, and there was no significant difference between the different soil layers (P > 0.05). The clay content was low, and the soil texture was sandy soil (international classification). Of the soil depths, the 0–10 cm soil layer had the highest sand content, which decreased with increasing soil depth. The soil water content of all soil profiles ranged from 4.14% to 7.42%, and the average values in the 0–60 cm soil layer for the three ground fissures were 5.03%, 5.02% and 6.07%. The soil saturated hydraulic conductivity decreased with increasing soil depth. The soil saturated hydraulic conductivity in the 0–10 cm soil layer was 1.10–1.19 mm min−1, and that in the 50–60 cm soil layer was 0.18–0.42 mm min−1, accounting for 15.25–38.18% of the 0–10 cm soil layer. For GF I, GF II and GF III, the average values of soil saturated hydraulic conductivity were 0.83, 0.64 and 0.83 mm min−1, respectively. Compared to that of GF II, the soil saturated hydraulic conductivities of GF I and GF III were slightly lower; however, there was no significant difference among the three ground fissures (P > 0.05). The soil organic matter decreased first, then increased and then decreased with increasing soil depth, which was distributed as an “S” shape. The soil organic matter in the 0–10 cm soil layer was the highest because of decomposition and migration of the litter.

Table 5 Soil physiochemical properties in the dump.

Plots	Soil depth
(cm)	SWC
(%)	SBD
(g cm−3)	CP
(%)	NPC
(%)	SHC
(mm min−1)	SOM
(g kg−1)	Soil mechanical composition (%)	
Sand	Silt	Clay	
GF I	0–10	6.03	1.30	36.70	14.27	1.10	32.66	77.21	12.25	11.54	
10–20	6.36	1.41	29.28	18.11	0.94	26.36	74.68	16.54	8.78	
20–30	5.09	1.41	30.42	16.95	0.87	21.47	76.58	10.25	13.17	
30–40	4.26	1.36	33.17	15.76	0.90	26.97	70.24	16.38	13.38	
40–50	4.31	1.34	32.72	17.17	0.73	20.64	71.53	13.84	14.63	
50–60	4.14	1.34	32.70	17.14	0.42	18.54	71.48	14.53	13.99	
0–60	5.03a	1.36a	32.50a	16.57a	0.83a	24.44a	73.62a	13.97a	12.58a	
GF II	0–10	7.16	1.31	33.41	17.35	1.18	22.82	78.62	12.05	9.33	
10–20	4.71	1.45	19.88	26.14	1.05	17.51	75.64	17.64	6.72	
20–30	4.30	1.31	34.49	16.27	0.73	18.56	75.34	16.37	8.29	
30–40	4.29	1.25	36.51	16.19	0.40	22.74	72.18	15.38	12.44	
40–50	4.68	1.22	38.29	15.35	0.27	23.82	70.39	16.08	13.53	
50–60	5.00	1.23	39.30	14.20	0.18	20.87	72.35	14.38	13.27	
0–60	5.02a	1.30a	33.65a	17.58a	0.64a	21.05a	74.09a	15.32a	10.60a	
GF III	0–10	6.04	1.39	29.55	18.40	1.19	29.77	74.35	15.34	10.31	
10–20	7.42	1.42	29.12	17.97	1.06	28.67	70.84	14.57	14.59	
20–30	4.89	1.43	27.91	19.03	1.25	25.62	69.58	16.13	14.29	
30–40	5.99	1.26	37.18	15.07	0.84	21.65	74.25	14.47	11.28	
40–50	6.05	1.31	36.19	14.39	0.40	24.04	73.35	14.28	12.37	
50–60	6.02	1.29	36.10	15.17	0.22	13.03	71.57	14.58	13.85	
0–60	6.07a	1.35a	32.68a	16.67a	0.83a	23.80a	72.32a	14.90a	12.78a	
Note:

The data in the table is the average. Different lowercase letters indicate significant differences among the three ground fissures (P < 0.05). SWC, soil water content; SBD, soil bulk density; CP, capillary porosity; NCP, non-capillary porosity; SHC, saturated hydraulic conductivity; SOM, soil organic matter; sand (0.02–2 mm), silt (0.02–0.002 mm), clay (<0.002 mm).

As shown in Table 6, the stained area ratio was significantly positively correlated with soil saturated hydraulic conductivity (P < 0.01), soil organic matter (P < 0.05) and sand content (P < 0.05) and was significantly negatively correlated with soil water content (P < 0.05) and clay content (P < 0.05); these results may be because the increase in soil saturated hydraulic conductivity improved soil hydraulic conductivity, increased the stained area and path of preferential flow, and then promoted the development of preferential flow. The increase in soil organic matter could have promoted soil aggregate formation and improved soil structure and soil infiltration capacity. The stained path number was significantly positively correlated with soil saturated hydraulic conductivity (P < 0.01) and soil organic matter (P < 0.05) and was not significantly correlated with the other indicators (P > 0.05). The stained path width had a significant positive correlation with soil saturated hydraulic conductivity, soil organic matter and sand content (P < 0.05) and a significant negative correlation with clay content (P < 0.05).

Table 6 Relationship between soil preferential flow and soil physicochemical properties.

Index	SAR	SPN	SPW	SWC	SBD	CP	NCP	SHC	SOM	Sand	Silt	Clay	
SAR	1.000	0.732**	0.981**	−0.548*	−0.192	−0.152	0.255	0.632**	0.527*	0.511*	−0.007	−0.527*	
SPN		1.000	0.647**	−0.461	−0.277	−0.257	0.383	0.746**	0.507*	0.324	0.199	−0.465	
SPW			1.000	−0.463	−0.176	−0.129	0.222	0.588*	0.544*	0.487*	−0.009	−0.505*	
SWC				1.000	0.154	−0.121	0.082	0.384	0.451	0.378	−0.335	−0.201	
SBD					1.000	−0.969**	0.815**	−0.688**	0.168	0.059	0.334	−0.092	
CP						1.000	−0.909**	−0.648**	−0.079	−0.018	−0.356	0.049	
NCP							1.000	0.654**	0.053	0.050	0.439	−0.162	
SHC								1.000	0.567*	0.199	0.193	−0.293	
SOM									1.000	−0.140	0.146	0.073	
Sand										1.000	−0.237	−0.777**	
Silt											1.000	−0.337	
Clay												1.000	
Notes:

* P < 0.05.

** P < 0.01.

n = 18.

Plant roots

As shown in Fig. 8, compared to GF I and GF III, the root density of GF II was greater. The root weight densities of the different soil layers of GF I, GF II and GF III were 0.453–0.938, 0.330–0.914, and 0.149–2.69 mg cm−3, respectively. With increasing soil depth, the root weight density of GF I and GF II first increased and then decreased, while that of GF III decreased. Compared to GF I and GF II, the root weight density of GF III in the 0–10 cm soil layer was 2.69 mg cm−3. The root length density in different soil layers of GF I ranged from 3.242 cm cm−3 to 6.243 cm cm−3, that of GF II ranged from 3.811 cm cm−3 to 7.744 cm cm−3, and that of GF III ranged from 3.906 cm cm−3 to 6.353 cm cm−3. The variation characteristics of root length density for the three fissures were the same as that of root weight density. The root surface area density in different soil layers of GF I ranged from 1.502 cm2 cm−3 to 2.532 cm2 cm−3, that of GF II ranged from 1.906 cm2 cm−3 to 2.647 cm2 cm−3, and that of GF III ranged from 1.844 cm2 cm−3 to 2.600 cm2 cm−3. With increasing soil depth, the root surface area density of GF I increased first and then decreased, while that of GF II and GF III decreased. The root surface area density of the three ground fissures was the highest in the 0–10 cm or 10–20 cm soil layer. The root volume density in different soil layers of GF I ranged from 0.056 cm3 cm−3 to 0.098 cm3 cm−3, that of GF II ranged from 0.061 cm3 cm−3 to 0.092 cm3 cm−3, and that of GF III ranged from 0.063 cm3 cm−3 to 0.087 cm3 cm−3. The variation characteristics of root volume density for the three fissures were the same as those of root surface area density.

Figure 8 Characteristics of plant root in the dump.

(A) The root density. (B) The root weight density. (C) The root length density. (D) The root surface area density. (E) The root volume density. The red columns represent GF I, the green columns represent GF II, and the blue columns represent GF III.

As shown in Table 7, the stained area ratio, stained path number and width were significantly positively correlated with plant roots and could promote the formation and development of preferential flow. The stained area ratio was significantly positively correlated with root length density (P < 0.01), root surface area density (P < 0.01) and root density (P < 0.05). The stained path number was significantly positively correlated with root length density (P < 0.05) but not with other indicators (P > 0.05). The stained path width was significantly positively correlated with root length density (P < 0.01), root surface area density (P < 0.01), root density (P < 0.05), and root weight density (P < 0.05) and was not significantly correlated with other indicators (P > 0.05).

Table 7 Relationship between soil preferential flow and plant root.

Index	SAR	SPN	SPW	RD	RWD	RLD	RSAD	RVD	
SAR	1.000	0.732**	0.981**	0.550*	0.426	0.664**	0.612**	0.338	
SPN		1.000	0.647**	0.280	0.203	0.505*	0.445	0.017	
SPW			1.000	0.589*	0.482*	0.639**	0.626**	0.343	
RD				1.000	0.573*	0.717**	0.692**	0.117	
RWD					1.000	0.692**	0.853**	0.467	
RLD						1.000	0.903**	0.307	
RSAD							1.000	0.472*	
RVD								1.000	
Notes:

* P < 0.05.

** P < 0.01.

n = 18. RD, root density; RWD, root weight density; RLD, root length density; RSAD, root surface area density; RVD, root volume density.

Discussion

Morphological characteristics of ground fissures

Ground fissures formed by natural soils, such as paddy soil, expansive soil and clay, are affected by soil texture (Gray & Allbrook, 2002), soil alternating drying and wetting (Zhang et al., 2013), freeze and thaw action (Maloof, Kellogg & Anders, 2002), and tillage methods (Bandyopadhyay et al., 2003). However, the formation and development of ground fissures in a dump is also mainly affected by the gravity of the soil, uneven subsidence and rainfall characteristics (Li et al., 2018; Zhang et al., 2012; Zhou et al., 2011). Therefore, the formation mechanism of ground fissures in a dump is different from that of natural ground fissures. The morphological characteristics of ground fissures in a dump are also unique, and ground fissures are distributed in front of the platform of a dump; most are linear, arcing and polygonal. Similar conclusions were drawn by Zhou et al. (2011). Ground fissures have no obvious frame structures. The numbers of fissures, fissure blocks, and nodes are relatively small. The depths of the fissures are also different. Ground fissures have relatively strong spatial variability yet clear self-similarity (Díaz-Fernández, Álvarez-Fernández & Álvarez-Vigil, 2010; Li et al., 2018). Studies have shown that ground fissures are distributed on different dump staircases, stretching along the contour lines. The widths of ground fissures vary, with the largest width reaching 10 cm (Hu & Mu, 2008).

Effect of ground fissures on preferential flow

Several important processes, such as infiltration, water storage in depressions, erosion and sedimentation, depend on soil microtopography (Bogner et al., 2012). Ground fissures not only change the soil microtopography and slope of the underlying surface but also destroy the continuity of the soil, thus accelerating the convergence of surface runoff, improving the efficiency of the catchment, and forming a preferential channel for water movement. Whether surface runoff occurs depends on the spray intensity and infiltration capacity. When the spraying intensity is greater than the infiltration capacity of the soil, the accumulated water continuously collects to form surface runoff; after the surface runoff flows into the ground fissures, it quickly moves downwards without being absorbed by the surrounding soil. The surface runoff continues to collect and fill in the ground fissures and then pass through the ground fissure, soil on both sides and bottom infiltrates into the soil. The depth of soil preferential flow can be reflected by the depth of soil matrix flow, which is the critical depth when the soil matrix flow turns to soil preferential flow, that is, the depth of the soil layer where the soil preferential flow stained area ratio is more than 80% (Bargués Tobella, Reese & Almaw, 2014). Bogner et al. (2012) concluded that the top 5 cm of soil is homogenously stained, indicating the dominance of matrix flow. There were narrow stained objects ranging from 5 cm to the bottom of the soil profiles. Guo et al. (2018) revealed that ground fissures in coal mining subsidence areas cause preferential flow. The width and number of fissures had a significant influence on the preferential flow in the coal mining subsidence area. The results of this study showed that the stained area of preferential flow was mainly located on both sides of the fissure, and flow morphological characteristics of the preferential flow were “above wide and below narrow”. The ground fissure width and width-to-depth ratio significantly promoted the development of preferential flow, and then, the development of preferential flow further intensified the expansion of ground fissures.

Effect of plant roots on preferential flow

The promotion of preferential flow by plant roots is a common result of multiple factors. Plant roots not only form a preferential path for soil water movement (Van Noordwijk, Schoonderbeek & Kooistra, 1993; Jiang et al., 2018) and increase soil pore connectivity (Cannavo & Michel, 2013) but also network soil and improve soil structure (Lin et al., 2010). Clark & Zipper (2016) noted that soil preferential flow of a dump is mainly concentrated near plant roots. The stained area ratio of a reclaimed forest was larger than that of reclaimed grasses, and the stained path width of a reclaimed forest was smaller than that of reclaimed grasses. An increase in plant roots can effectively increase the stained area and stained path width. In addition, an effective increase in root length will increase not only root expansion in the horizontal direction but also root extension in the vertical direction. Root surface area can increase the contact area between roots and soil and form a preferential pathway channel on the root surface, which promotes the development of soil preferential flow and accelerates water movement (Bogner et al., 2010; JoRgensen et al., 2002). The denser roots of the plant, the larger the contact area with the soil. Shao et al. (2020) pointed out that fine roots can significantly promote the formation of preferential soil flow. However, the stained area ratio, stained path number and width were positively correlated with root volume density but not significantly (P > 0.05). This was because the root volume was an index of three-dimensional space, while the stained area ratio, stained path number and width were the indexes of two-dimensional space, which was the projection of stained area on the plane. Therefore, the root volume density did not play a decisive role in preferential flow.

Conclusions

The three ground fissures had different degrees of curvature, and the fissure width was significantly different (P < 0.05), while the fissure depth was not significantly different. The ground fissure width-to-depth ratio decreased with increasing soil depth and then stabilized. There was no significant difference in preferential flow values among the three fissures, and the values were distributed as a “T” shape. The stained area was greater than 90% in the 0–5 cm soil layer for the three fissures. The stained area was evenly distributed, preferential flow development was not obvious, and the water movement was dominated by matrix flow. The stained area decreased below the 10 cm soil layer, and the development degree of preferential flow was obvious and was mainly concentrated on both sides of the fissure. The stained area ratios of GF I, GF II and GF III were 27.23%, 31.97%, and 30.73%, respectively, and these ratios decreased with increasing soil depth and were distributed as an “S” shape. The maximum stained area ratio was 90.70% in the 0–10 cm soil layer, and the minimum was 62.51%. The stained depth of GF II was significantly greater than that of GF I and GF III. The spatial variability in the stained area ratios for the three fissures was moderate. With increasing soil depth, the stained path number first increased and then decreased. The stained path width of the surface soil reached 90 cm, and the stained path width decreased with increasing soil depth and then finally trended towards 0. Soil preferential flow was affected by both fissure width and depth. The stained area ratio, stained path number and width were significantly positively correlated with soil saturated hydraulic conductivity, soil organic matter, sand content and plant roots, and these indicators could promote the formation and development of preferential flow.

Supplemental Information

Supplemental Information 1 Raw data.

OPJ files can be opened with Origin Graph 9.1.

Click here for additional data file.

Supplemental Information 2 Raw numerical data for Tables 2, 4, 6, and 7.

Click here for additional data file.

Additional Information and Declarations

Competing Interests

Author Contributions

Data Availability

The authors declare that they have no competing interests.

Yexin Li conceived and designed the experiments, authored or reviewed drafts of the paper, and approved the final draft.

Gang Lv conceived and designed the experiments, authored or reviewed drafts of the paper, and approved the final draft.

Hongbo Shao conceived and designed the experiments, authored or reviewed drafts of the paper, and approved the final draft.

Quanhou Dai analyzed the data, prepared figures and/or tables, and approved the final draft.

Xinpeng Du performed the experiments, prepared figures and/or tables, and approved the final draft.

Dong Liang performed the experiments, prepared figures and/or tables, and approved the final draft.

Shaoping Kuang analyzed the data, prepared figures and/or tables, and approved the final draft.

Daohan Wang analyzed the data, authored or reviewed drafts of the paper, and approved the final draft.

The following information was supplied regarding data availability:

Raw data and image files are available in the Supplemental Files.

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
