# Peer review of "Determining the influencing factors of preferential flow in ground fissures for coal mine dump eco-engineering"

_PeerJ, doi:10.7717/peerj.10547_

## Round 0.1 · original submission · Minor Revisions

Two reviewers have provided constructive comments and suggestions. Please address these comments point-by-point in your response, indicating how and where the manuscript was modified. Please pay particular attention to the questions/comments from both reviewers regarding how and why three ground fissure examples were selected from the larger group, and whether/how these are representative.

Reviewer 1 ·

Basic reporting

In this manuscript, the Authors investigated the morphological characteristics of preferential flow. The overall language of the manuscript is Ok, proofreading by a native speaker will further improve the quality of the manuscript. Language issues have been observed regarding the correct usage of articles and abbreviations e.g. “GF” in line 44. In Line 105, from “Water flows” I presume you mean “water flow in soil”. The introduction is well written, however, in the last paragraph adding few studies regarding ground fissures in the dump area will further improve this section. All the Figures and Tables are relevant and well presented.

Experimental design

The topic of the manuscript fits well in the Journal and its objectives are of great interest to readers. The experimental design is easy to follow and the study can be helpful to understand the groundwater movement in mining areas. The study site description needs more details and it must be revised. In the study site description, the location map as shown in Figure 1 does not provide sufficient information regarding the study area it is better to pin this location at the province/country map. The climate conditions have been described in detail but it's not clear what is the main source of this information, how many and which years data have been used to get these averages?. It is better to add a flow chart of your methodology. There are 61 ground fissures in total why only 3 have been selected? In section 2.2.2 Line 219-221 add some references for the drying method and ring method. Please elaborate on which type of statistical analysis has been performed and how you dealt uncertainties involved during the image processing.

Validity of the findings

The results are well presented and support the conclusions. The discussion is sound and relevant, the literature cited in the introduction is nicely used for discussion. The results will be more meaningful if Authors further elaborate regarding:
The effects of soil macropores on effluent rate and hydraulic conductivity.
The relationship between the dye coverage ratio and plant factors

Additional comments

I enjoyed reading this manuscript which is scientifically sound and the findings are interesting.

Reviewer 2 ·

Basic reporting

This paper revealed how preferential flow propagated as affected by natural ground fissures. The data were well analyzed, the contents of this paper are full and complete, the structure is reasonable. But some conclusions drawn from this paper need further consideration.

Experimental design

1. there were no treatments. The author just selected some experimental points and I wonder how representative these three situ points are.
2. the author just selected one sprinkling intensity, i.e. 50 mm/h. I think this intensity could indirectly affect the preferential regime, including the stained area, strained patterns, infiltration depth, etc. I suggest the author carefully elucidate based on what this intensity was chosen.

Validity of the findings

1. the scientific problems are not quite clear because of the poor introduction. I think this scientific research is doable but I cannot find the necessity in the introduction section.
2. the thresholding process from color image to binary image determined the infiltration path. I suggest the author explain how you overcome the differences of illumination and photographing time points, homogeneity of staining when the stained images were photographed and binarized.

Additional comments

Line 44: what’s the definition of ‘The preferential flow stained area ratios’. Does it make sense when you chose different size of research area?
Line 54: how can you determine the 3-dimensional water content prior to the experiments?
Line 103~111: the authors did not specify the necessity of research and the scientific problems. The research background was not highly concentrated on the research problem. Such as, until Line 103~111, the author still introduce the research background regarding crack flow, rather than research progress.
Line 112~117: the same problem with Line 103~111, the research progress is not clear, especially on the phenomenon of preferential flow.
The whole introduction failed to introduce recent research progress on fissure preferential flow and did not put forward a clear research problem regarding preferential flow in ground fissures. I think this part should be thoroughly enhanced.
Line 255: the parameter ‘stained area ratio’ is inappropriate here because it is totally determined by the depth of the soil profile which is selected by the authors.
Line 461~476: the discussion on how ground fissures affected preferential flow should be strengthened, such as the whole hydrology process from sprinkling to the end of preferential flow; why the top 5 cm showed a matrix flow regime; the specific process of how preferential flow were “above wide and below narrow”, etc.
Line 487~489: I still cannot understand why the preferential flow will be promoted with the increase in the contact area between roots and soil. Please further clarify.

---

## Round 0.2 · Minor Revisions

Both previous reviews recommended minor revisions, and your explanations in the Revison Notes address those adequately in my opinion, so I did not send the manuscript for a second review. However, although your comments in the Revision Notes are very helpful, in several cases they were not fully reflected in the revised manuscript.

For example, your response in the Revision Notes to Reviewer 1 comment #2 and Reviewer 2 comment #2 about the selection of the three fissure is thorough and helpful, but in the revised manuscript itself you only added the words "by statistical analysis" (line 194). I think additional information as provided in the Revision Notes should be included in the manuscript, as the readers are likely to have the same question as the reviewers.

Similarly, in response to Reviewer 1, comment #2 you noted that the length of the meterological record was approximately 30 years, but again that information was not added to the revised manuscript.

In other cases, such as your response to Reviewer 2, comment #4, you have done a good job of incorporating the response into the manuscript (in this case, lines 265-268). This approach needs to be applied consistently across all the comments.

To summarize, please do not only respond to the reviewer comments in the Revision Notes, but also be sure to make corresponding modifications to the manuscript, as readers will likely have the same questions as the reviewers.

Please revise the manuscript again (minor revisions) to ensure that the responses given to the reviewers in the Revision Notes are fully reflected in the revised manuscript. In the response notes please indicate which lines in the revised manuscript have been updated to reflect the comments.

---

## Round 0.3 · accepted · Accept

Thank you for the additional revisions and clarifications. These will help readers to better understand and interpret your work.